# *In utero* exposure to protease inhibitor-based antiretroviral regimens delays growth and developmental milestones in mice

**Ambalika Sarkar**[1], **Kayode Balogun**[1], **Monica S. Guzman Lenis**[1], **Sebastian Acosta**[1], **Howard T. Mount**[2,3], **Lena Serghides**[1,3,4,5]*

**1** Toronto General Hospital Research Institute, University Health Network, Toronto, Ontario, Canada, **2** Departments of Psychiatry & Physiology, Tanz Centre for Research in Neurodegenerative Diseases, University of Toronto, Toronto, Ontario, Canada, **3** Institute of Medical Sciences, University of Toronto, Toronto, Ontario, Canada, **4** Women's College Research Institute, Toronto, Ontario, Canada, **5** Department of Immunology, University of Toronto, Toronto, Ontario, Canada

* lena.serghides@utoronto.ca

**Data Availability Statement:** All relevant data are within the paper and its Supporting Information files.

## Abstract

Antiretroviral therapy (ART) in pregnancy has dramatically reduced HIV vertical transmission rates. Consequently, there is a growing number of children that are HIV exposed uninfected (CHEUs). Studies suggest that CHEUs exposed *in utero* to ART may experience developmental delays compared to their peers. We investigated the effects of *in utero* ART exposure on perinatal neurodevelopment in mice, through assessment of developmental milestones. Developmental milestone tests (parallel to reflex testing in human infants) are reflective of brain maturity and useful in predicting later behavioral outcomes. We hypothesized that ART in pregnancy alters the *in utero* environment and thereby alters developmental milestone outcomes in pups. Throughout pregnancy, dams were treated with boosted-atazanavir combined with either abacavir/lamivudine (ATV/r/ABC/3TC), or tenofovir/emtricitabine (ATV/r/TDF/FTC), or water as control. Pups were assessed daily for general somatic growth and on a battery of tests for primitive reflexes including surface-righting, negative-geotaxis, cliff-aversion, rooting, ear-twitch, auditory-reflex, forelimb-grasp, air-righting, behaviors in the neonatal open field, and olfactory test. *In utero* exposure to either ART regimen delayed somatic growth in offspring and evoked significant delays in the development of negative geotaxis, cliff-aversion, and ear-twitch reflexes. Exposure to ATV/r/ABC/3TC was also associated with olfactory deficits in male and forelimb grasp deficits in female pups. To explore whether delays persisted into adulthood we assessed performance in the open field test. We observed no significant differences between treatment arm for males. In females, ATV/r/TDF/FTC exposure was associated with lower total distance travelled and less ambulatory time in the centre, while ATV/r/ABC/3TC exposure was associated with higher resting times compared to controls. *In utero* PI-based ART exposure delays the appearance of primitive reflexes that involve vestibular and sensory-motor pathways in a mouse model. Our findings suggest that ART could be disrupting the normal progress/maturation of the underlying neurocircuits and encourage further investigation for underlying mechanisms.

**Funding:** This work was supported by the Canadian Foundation for AIDS Research (Grant 27-010) and the Canadian Institutes of Health Research (CIHR HIV/AIDS Comorbidities Prevention and Healthy Living –Team Grant) awarded to LS. KB was supported by a CIHR Canadian HIV Trials Network fellowship.

**Competing interests:** The authors have no conflicts of interest to disclose in relation to this work. LS reports personal fees from ViiV Healthcare for participation in a Women and Transgender Think Tank. This does not alter our adherence to PLOS ONE policies on sharing data and materials.

**Abbreviations:** 3TC, lamivudine; ABC, abacavir; ART, antiretroviral therapy; ATV/r, ritonavir-boosted atazanavir; CHEUs, children that are HIV-exposed uninfected; FTC, emtricitabine; GD, gestational day; GW, gestational weeks; NRTI, nucleoside reverse transcriptase inhibitor; P1, postnatal day 1; PI, protease inhibitor; TDF, tenofovir; U-ART, *in utero* antiretroviral therapy.

# Introduction

The use of ART in pregnancy has enabled a radical reduction in HIV vertical transmission [1]. Consequently, there is now a large and rapidly increasing number of children that are HIV-exposed uninfected (CHEUs). The number of CHEUs exposed *in utero* to ART is estimated at over 10 million globally and is increasing by 1.5 million each year [2]. In Canada, CHEUs under 12 years of age number at 2,500 and the population is growing by 200–300 annually [2]. Although most CHEUs are in good health, concerns remain about the effects of *in utero* exposure to ARTs for which, pregnancy-relevant safety/toxicity data are limited. Observational cohort studies suggest that CHEUs may be at increased risk of adverse health outcomes including higher incidence of pre-term birth and growth restriction [3], poorer early-life height and weight growth outcomes [4], increased morbidity and mortality in the first year of life [5], increased susceptibility to infection [6, 7], language and cognitive delays [8, 9], retention of primitive reflexes [10], and other neurological impairments [11, 12]. An observed decline in IQ scores in CHEUs followed in a longitudinal study at 3.5 and 5.5 years of age suggested that some of the negative effects of ART exposure, may not be visible in infancy but may emerge over time [13]. Systematic reviewing of literature has identified delayed cognitive and executive developments in CHEUs [14, 15]. Delayed emergence of language, hearing, and lower scores in socioemotional domains on the Bayley Scale III have been observed in several studies, particularly in CHEUs exposed to protease inhibitor (PI)-based regimens [11, 12, 16]. Investigating neurocognitive outcomes in CHEUs is challenging and results can be influenced by confounding factors such like HIV disease severity, maternal mental health, and education level, family environment and socioeconomic status [15, 17]. In fact, methodological shortcomings such as absence of appropriate control groups in many studies have rendered them inconclusive. However, after careful adjustment for confounding factors, evidence suggests that *in utero* ART (U-ART) exposure comprising of various classes of antiretrovirals are capable of delaying growth and neurocognitive development in CHEUs.

Delayed emergence of neurocognitive abilities generally indicates delayed growth/maturation of the underlying neurocircuits [18]. For early prediction of disabilities that may emerge later in life, it is important first and foremost to identify delays in the development of postnatal milestones in newborns exposed to ART *in utero*. We hypothesized that ART in pregnancy alters the in-utero environment for the developing fetal brain, which in turn leads to altered developmental milestone outcomes in neonates. To test this hypothesis we used a mouse model of *in utero* ART exposure and examined the effects of two nucleoside reverse transcriptase inhibitor (NRTI) backbones abacavir plus lamivudine (ABC/3TC) and tenofovir disoproxil fumarate plus emtricitabine (TDF/FTC) in combination with the ritonavir-boosted PI atazanavir (ATV/r), on somatic growth and primitive reflexes in mouse pups. Both these combination regimens are recommended for use in pregnancy, and ATV/r exposure has been linked to delayed language and socio-emotional development in CHEU cohort studies [11, 12, 16]. Using a mouse-model allowed us to examine the effects of U-ART in offspring in the absence of maternal, environmental, and HIV-related confounders.

# Materials and methods

## Animals

C57BL/6J mice, 10 males, 50 females (Jackson Laboratories, USA) were used for breeding at 10 weeks of age (1 male × 2 females, per cage). Animals were maintained under standard laboratory conditions with ad-libitum access to food and water and 12 h light/dark cycle. All animal experiments were performed with the approval of the Toronto Centre for Phenogenomics

(TCP) Animal Care Committee, based on national guidelines of the Canadian Council for Animal Care (CCAC).

## Treatment of the dams

Female mice were trained for oral gavage (feeding needle size: 20Gx38 mm) with water for a week prior to mating, to avoid losing pregnancy from gavage related stress. The morning after mating (gestational day (GD) 0.5), plugged females were separated from males, randomly assigned to one of the two treatment arms and administered either ABC/3TC+ATV/r (100/50mg/kg/day + 50/16.6mg/kg/day) or TDF/FTC+ATV/r (33.3/50mg/kg/day + 50/16.6mg/kg/day) by oral gavage, or control which received an equal volume of water. Drugs were acquired by prescription, crushed, and suspended in sterile water. Dosing was determined through previous experiments [19] and is known to lead to peripheral drug levels equivalent to those seen in pregnant women. On GD18, pregnant dams were separated into individual cages where they delivered their litters. Upon delivery of litters (Postnatal day 0), dams stopped receiving treatment. On P1 large litters were culled down to 6 pups/litter in order to maintain uniformity in handling time during testing.

## Pup handling

From postnatal day 1 (P1) onwards, pups were assessed daily for appearance of developmental milestones as described by Hill et al, 2008 [20]. Dams were gently removed from their home cages and held in a separate room while each litter was tested. Litters were removed from the cages and placed in weighing boats, lined with tissue papers and were kept on heating pads to ensure that pups retained their body temperatures. Pups in each litter were individually weighed and measured for body length and visually examined for somatic developments: appearance of fur, teeth, opening of eyes and pinna detachment. Each day (between 10:00–15:00hrs), each pup was evaluated on a battery of tests that assessed reflex development and attainment of milestones that were appropriate for their age (S1 Fig), by individuals blind to treatment groups, working in pairs. After testing, pups were returned to their nests and reunited with the dams. A total of 145 pups (83 males and 62 females) from 25 different litters were tested for developmental milestones. Pups were allowed to grow into adulthood after completion of the milestones study. At 2 months of age, animals were tested for activity in the open field test (OFT).

## Developmental milestone procedure

Surface righting (P1-P13): Pups were individually held on their backs (supine position), with paws facing upwards. Time taken by pups to flip over onto their abdomens (acquire prone position) were noted. Test was terminated if the pup failed to turn over within 30 seconds. Surface righting was measured once daily until the pup could right itself in less than 1second for two consecutive days. A mature response is expected between P8–P10 [21]. The human equivalent of this test is the tonic labyrinthine reflex seen in babies between 0–6 months of age [22].

Negative geotaxis (P1-P14): Pups were individually placed on a wire mesh set at a 45˚ angle with heads facing downwards. Time taken to turn around by 180˚ and move upwards was recorded. Test was repeated daily until pups passed in <30s for two consecutive days. Mature response in this test is expected by P3–P5 [20].

Cliff aversion (P1-P14): Pups were positioned on the edge of a small box with forepaws and the snout hanging over the edge. Time taken to turn and crawl away from the edge was noted. Test was repeated daily until pups passed in <30s for two consecutive days. Mature response expected by P4–P5 [20, 23].

Rooting (P1-P12): Pups were gently stroked with the twisted filament of a cotton tip applicator along the side of the head. A positive response was noted if the pup moved its head toward the filament. Test was repeated daily until pups responded correctly for two consecutive days. Mature response in this test is expected by P4–P5 [20, 21]. The human equivalent of this is the rooting reflex (for suckling) observed in babies between 0–4 months of age [24, 25]

Forelimb grasp (P4-P14): Pups were held with their forelimbs grasping a wire bar, while hindlimbs were not in touch with any surface. Length of time pups were able to hold onto the bar and remain suspended was noted. Test was repeated daily until pups were able to stay suspended for a minimum of 1s for two consecutive days. Mature response expected by P8-P9 [20, 21]. The human equivalent for this is the palmar grasp reflex observed in babies between 0–5 months of age [25, 26]

Auditory startle (P7-P18): Pups were individually exposed to an acoustic stimulus (hand clapping, 10 inches away from the pup). The first day on which pups responded with a quick involuntary jump was noted. Mature response expected by P10-P14 [20]. The humans, startle reflexes are observed between 0–5 months of age [26].

Ear twitch (P7-P15): The fine filament of a cotton tip applicator is gently brushed against the tip of the pup's ear. Positive responses were noted if pups responded by flattening their ears. Test was repeated daily until pups responded correctly for two consecutive days. Mature response expected by P9–P10 [20].

Open field traversal (P8-P21): Tests straight line walking in pups, as opposed to pivoting. Each pup was placed at the centre of a circular arena (diameter = 13cm). Latency to move out of the arena was recorded. Test was repeated daily until pups walked out of the arena in <30s for two consecutive days. A mature response in this test is expected by P10-P13 [27].

Air righting (P8-P21): Pups were held on their backs (with four paws turning upwards), 10cm above a cotton pad, and released. Reflex was considered mature when the pup turned its body during the fall and landed on its paws. Test was repeated daily until pups landed on all four paws for two consecutive days. Mature response in this test is expected by P10-P12 [20].

Olfaction (the homing test): On P11, individual pups were transferred to a test cage layered two-third with clean and one-third with soiled bedding from the home-cage. The pup was placed in the middle of the chamber and allowed to wander. Homing was considered successful if the pup moved over and stayed in the portion of the cage that contained soiled home-cage bedding within 2.5 mins [21].

## Open field test in adulthood

Control and U-ART animals (n = 12–17 per group, per sex) were tested for general activity and anxiety-like behavior in the open field using Activity Monitor (Med Associates Inc, Fairfax, USA). Horizontal and vertical activity of mice were monitored by photo-beam channels in the open field (40 × 40 × 40 cm), for a period of 15 mins. A centre (20 x 20 x 20 cm) and a periphery were demarcated virtually within the open filed arena using the Activity Monitor software. Automated tracking was used to score time spent and distance moved in the centre and in the periphery of the arena, vertical (rearing) activity, resting time and speed of movement of the mice.

## Statistical analysis

An N = 83 male and N = 62 female pups from 6–10 different litters were assessed per arm for all developmental milestone tests. Data are presented as means with standard errors. All analyses were performed stratified by sex. Using a mixed-effects model that included treatment as a fixed effect and litter as a random effect, we explored associations between developmental outcomes and treatment accounting for litter effects. Differences in weight and length were

assessed for each day, stratified by sex, accounting for litter effects as above. Differences in the homing test were examined using chi-square test. An N = 12–17 per sex per arm were used in the OFT. Statistical difference in the OFT were assessed using one way ANOVA with Bonferroni multiple comparison test. A p<0.05 was deemed significant. STATA v.13.0 (StataCorp Ltd., USA) and Prism v.5 were used for all analyses. All raw data are provided in S1 Data.

## Results

We assessed pups daily for physical growth parameters and on a battery of tests for primitive reflexes including surface-righting, negative-geotaxis, cliff-aversion, rooting, ear-twitch, auditory-reflex, forelimb-grasp and air-righting, and for behaviors in the neonatal open field, and olfactory test. Given the fact that key developmental milestones are conserved between different mammalian species [28, 29], we have provided a comparison of the major neurodevelopmental events in mice and humans in Fig 1 along with the timeline of our treatments and tests, for better contextualization of the study presented here.

### Mortality

Since increased mortality in the first year of life has been observed with CHEUs we first examined pup survival in the post-natal period [2]. We observed no mortality in the control group, with all pups surviving for the entirety of the experiment. In the ATV/r/ABC/3TC group we observed a 3% mortality rate (2 of 66 pups, 2 of 9 litters with at least one mortality) and in the ATV/r/TDF/FTC groups we observed a 6.4% mortality rate (5 of 78 pups, 4 of 10 litters with at least one mortality). The mortality rate did not differ significantly between groups (p = 0.14 by $\chi 2$ test). All pup death occurred on P0, with no pup loss observed after P0 in any group.

### Somatic development

Somatic development was assessed in control and U-ART pups by measuring body weights, body lengths, and visual inspection for detachment of pinna, the appearance of fur, opening of eyes, and eruption of incisors.

Pups exposed *in utero* to ATV/r/ABC/3TC and ATV/r/TDF/FTC showed significantly lower birth weights as compared to controls (Fig 2A–2C). Weights of male pups exposed to ATV/r/ABC/3TC were lower than that of controls on P1 (p = 0.002). Similarly, weights of male pups exposed to ATV/r/TDF/FTC were significantly lower than that of controls on P1 (p = 0.001) and P2 (p = 0.019) (Fig 2A). Weights of female pups exposed to ATV/r/ABC/3TC or ATV/r/TDF/FTC were significantly lower than that of controls on P1 (p = 0.001, p<0.0001 respectively) and P2 (p = 0.05, p = 0.039 respectively) (Fig 2C). U-ART pup body weights in both sexes reached control levels by P3 (Fig 2B and 2D).

No remarkable difference in body length was observed between groups in male pups exposed to either treatment regimens (Fig 2E). Body lengths of female pups exposed to ATV/r/ABC/3TC were significantly lower than that of controls from P1 (p = 0.03) till P5 (p = 0.043) (Fig 2F). Body length did not differ in females exposed to ATV/r/TDF/FTC compared to control (Fig 2F).

In C57BL/6 mice, the expected timeline for detachment of pinna is P3-P4, for appearance of fur is P6-P7, for eruption of incisors is P10-P12 and for opening of the eyes is P11-P13 [30], and this coincided with findings in our control group. Delay in detachment of pinna was observed in the ATV/r/ABC/3TC exposed female pups (p = 0.004) and the ATV/r/TDF/FTC exposed male (p<0.0001) and female (p = 0.002) pups (Fig 3A). No significant differences were observed between treatment groups in appearance of fur (Fig 3B), opening of eyes (Fig 3C), and incisors eruption (Fig 3D).

**A.  In mice**

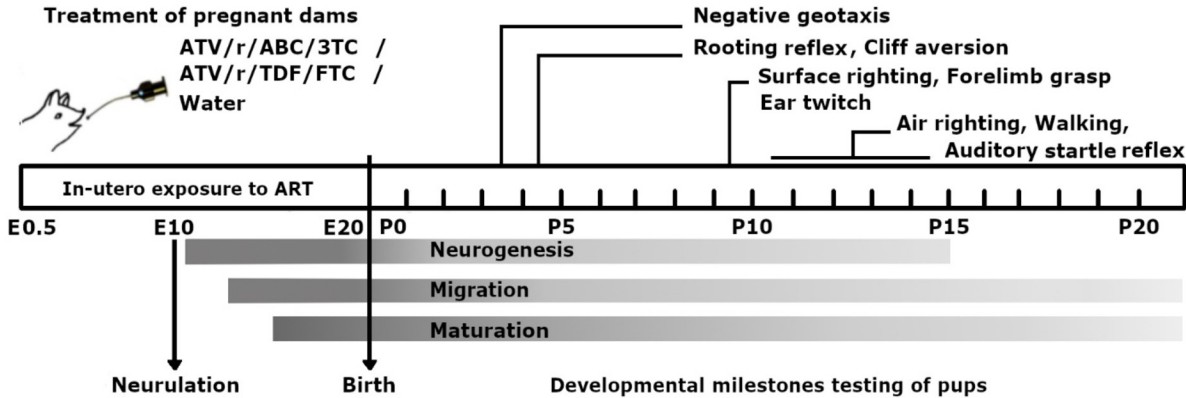

**B.  In humans**

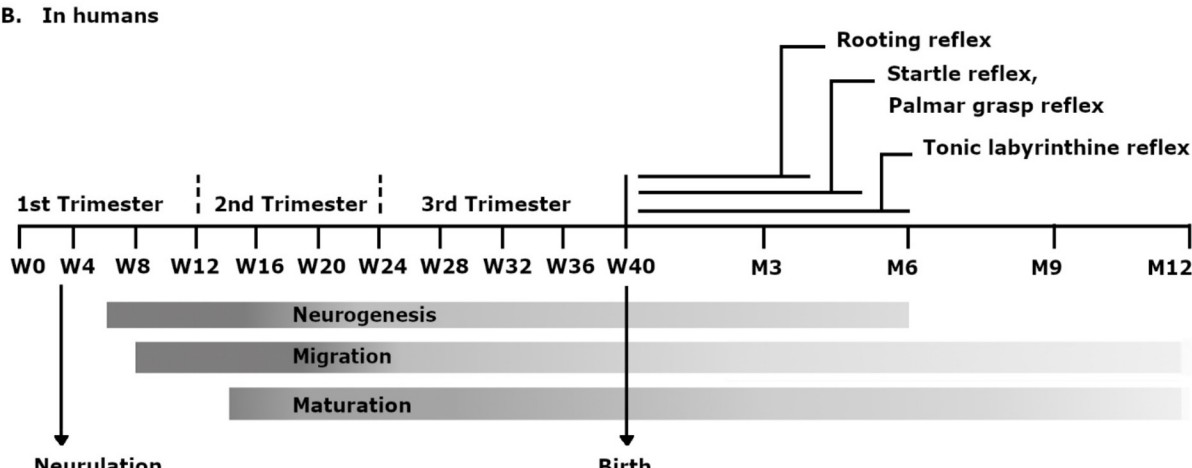

**Fig 1. Comparative timelines of neurodevelopment and reflex maturation in mice and humans.** Shown is the timeline for major neurodevelopmental events and reflex maturation in mice (A) and in humans (B) and a schematic representation for the experimental design in mice (A). Pregnant dams received ART from embryonic day 0.5 to E20 (till birth of pups). Developmental milestones were assessed in pups daily from P1 to P21. Being altricial in nature, a considerable amount of neurodevelopment occurs in mice pups postnatally. Timing of neurogenesis, maturation and migration of neuronal processes have been depicted using shaded rectangles for both species. Postnatal days on which mature responses are normally expected for specific reflex tests in mice have been shown with pointers in (A). Comparable reflexes in human infants and the timing for maturation and disappearance of these reflexes have been shown in (B).

Taken together, our results show that U-ART pups are born small for gestational age and have delayed somatic growth during the first two postnatal days. U-ART pups managed to catch-up with the controls in measures of somatic growth around P3. A slight delay in pinna detachment was observed but generally, somatic development milestones in U-ART pups were similar to controls.

## Development of primitive reflexes

We assessed the development of neurological reflexes in U-ART and control pups. *In utero* exposure to ATV/r/ABC/3TC and ATV/r/TDF/FTC delayed the emergence of negative-

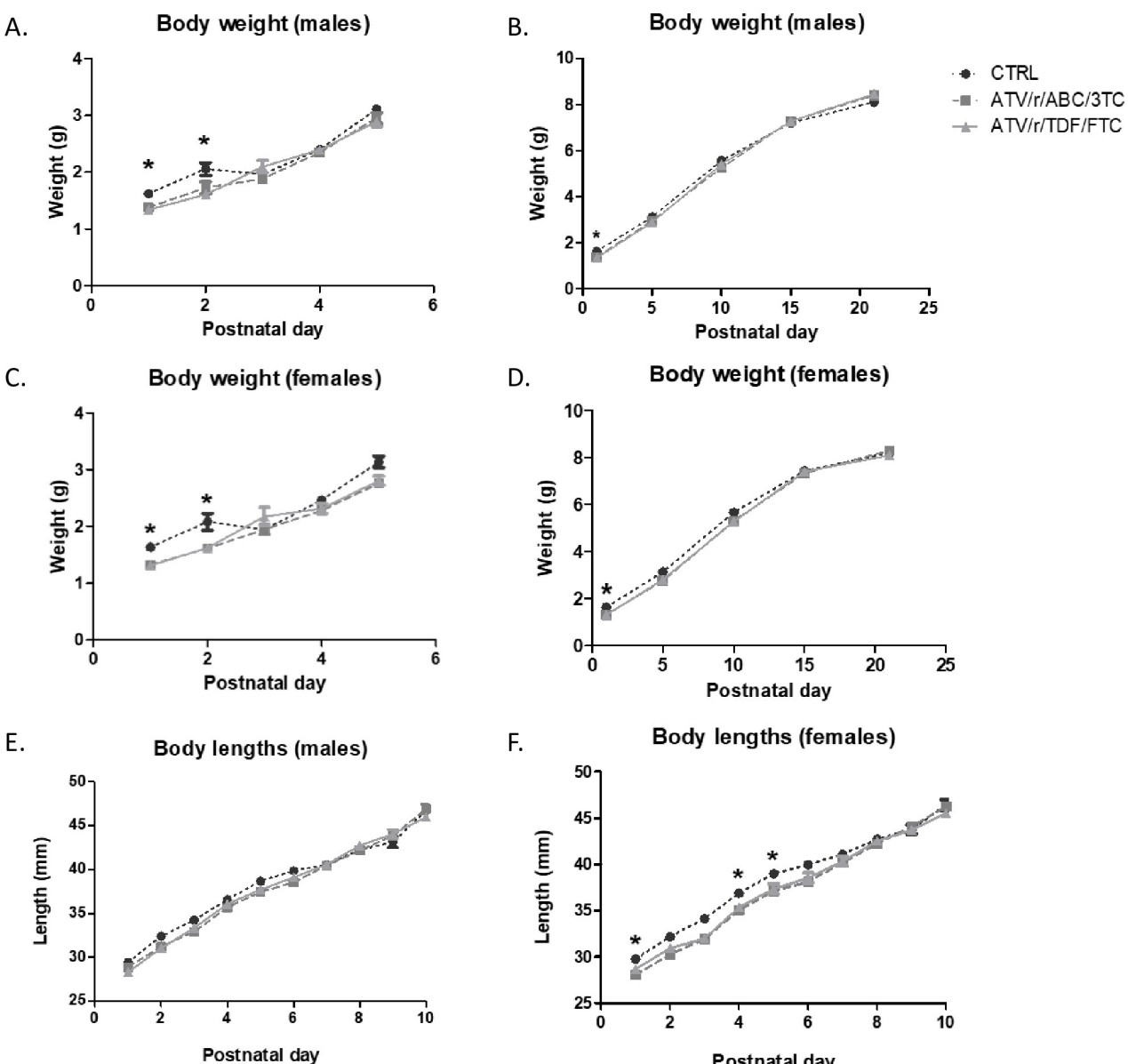

**Fig 2. Pups exposed in-utero to ART are born small and weigh less for gestational age.** Pup weight and length were assessed daily from postnatal day 1–21. Weight for the first 5 day is shown in (A) for males and (C) for females. Weight for the entire course is shown in (B) for males and (D) for females. Length is shown in (E) for males and (F) for females. Data are mean ± SEM (n = 11–34 pups/group). A mixed effects model was used to examine differences at each day between control and treatment arms (fixed effect) accounting for litter effects (random effect). *p<0.05 vs. control. ATV/r, ritonavir-boosted atazanavir; ABC, abacavir; 3TC, lamivudine; TDF, tenofovir; FTC, emtricitabine; CTRL, control.

geotaxis and cliff-aversion reflexes in male and female pups (Fig 4A and 4B). Mature responses are expected in these tests normally by P4 and P5 respectively [20, 23], as seen in our control group. In most U-ART pups, mature responses were not observed before P6. *In utero* exposure to ATV/r/ABC/3TC delayed emergence of negative-geotaxis in male pups approximately by 2 days (p = 0.01) and in females by 2.5 days (p = 0.001) (Fig 4A). Similarly, exposure to ATV/r/TDF/FTC delayed the appearance of negative-geotaxis in male pups approximately by 2 days (p = 0.008) and in females by 2.5 days (p = 0.001) (Fig 4A). *In utero* exposure to ATV/r/ABC/3TC delayed emergence of cliff-aversion in males (p = 0.012) and females (p = 0.004)

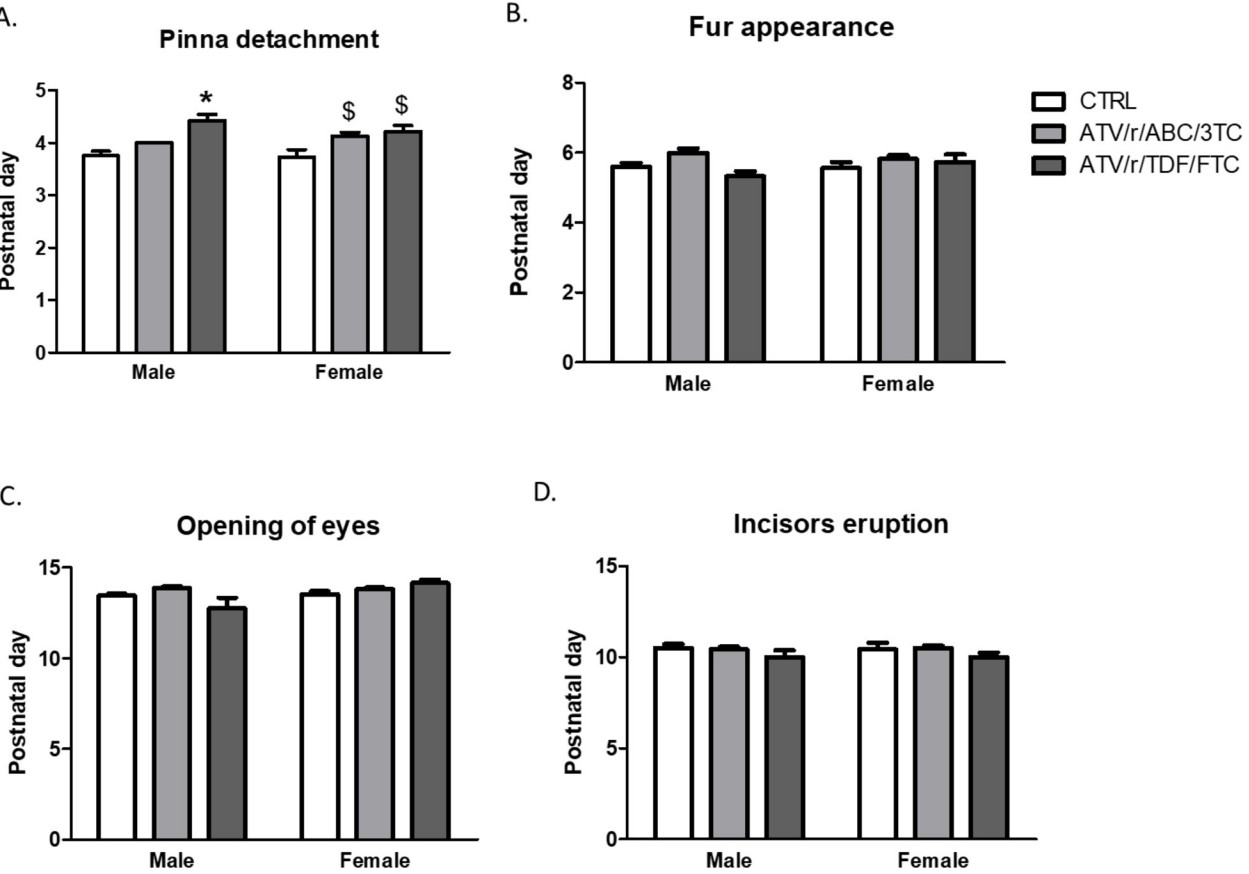

**Fig 3. In-utero exposure to ART delays somatic development.** Postnatal day of pinna detachment (A), fur appearance (B), eye opening (C), incisor eruption (D) for male and female pups exposed in-utero to control (CTRL, white bars), ATV/r/ABC/3TC (light grey bars), or ATV/r/TDF/FTC (dark grey bars). Data are mean ± SEM (n = 11–34 pups/group). A mixed effects model was used to examine differences between control and treatment arms (fixed effect) accounting for litter effects (random effect) stratified by sex. *p<0.05 compared to control males. $p<0.05 compared to control females. ATV/r, ritonavir-boosted atazanavir; ABC, abacavir; 3TC, lamivudine; TDF, tenofovir; FTC, emtricitabine; CTRL, control.

approximately by 2 days (Fig 4B). Similarly, *in utero* exposure to ATV/r/TDF/FTC delayed the appearance of cliff-aversion in both males (p = 0.012) and females (p = 0.004) approximately by 2 days (Fig 4B).

Development of rooting, a tactile reflex, was delayed selectively in the male pups exposed *in utero* to ATV/r/ABC/3TC (p = 0.023) or ATV/r/TDF/FTC (p = 0.017) approximately by a day (Fig 4C). In female pups, rooting was unaffected by exposure to either treatment regimens (Fig 4C).

Development of ear twitch, another tactile reflex, was also delayed in the male pups exposed to ATV/r/TDF/FTC (p = 0.001) and in females exposed to ATV/r/ABC/3TC (p = 0.028) as well as ATV/r/TDF/FTC (p = 0.001) (Fig 4D) by 1–1.5 days.

*In utero* exposure to ATV/r/ABC/3TC delayed the development of forelimb grasp reflex, selectively in females (p = 0.002) approximately by a day (Fig 4E). Forelimb grasp was not affected in males exposed to either treatment regimen.

Performance in the homing test, an olfaction test, was poorer than control in U-ART males, although only male pups exposed to ATV/r/ABC/3TC reached significance (p = 0.01) (Fig 4F).

A few primitive reflexes were unaltered by U-ART. No significant differences were noted between treatment groups for the appearance of surface-righting, auditory-reflex, air-righting, and straight-line walking, assessed by open field traversal test (Fig 5).

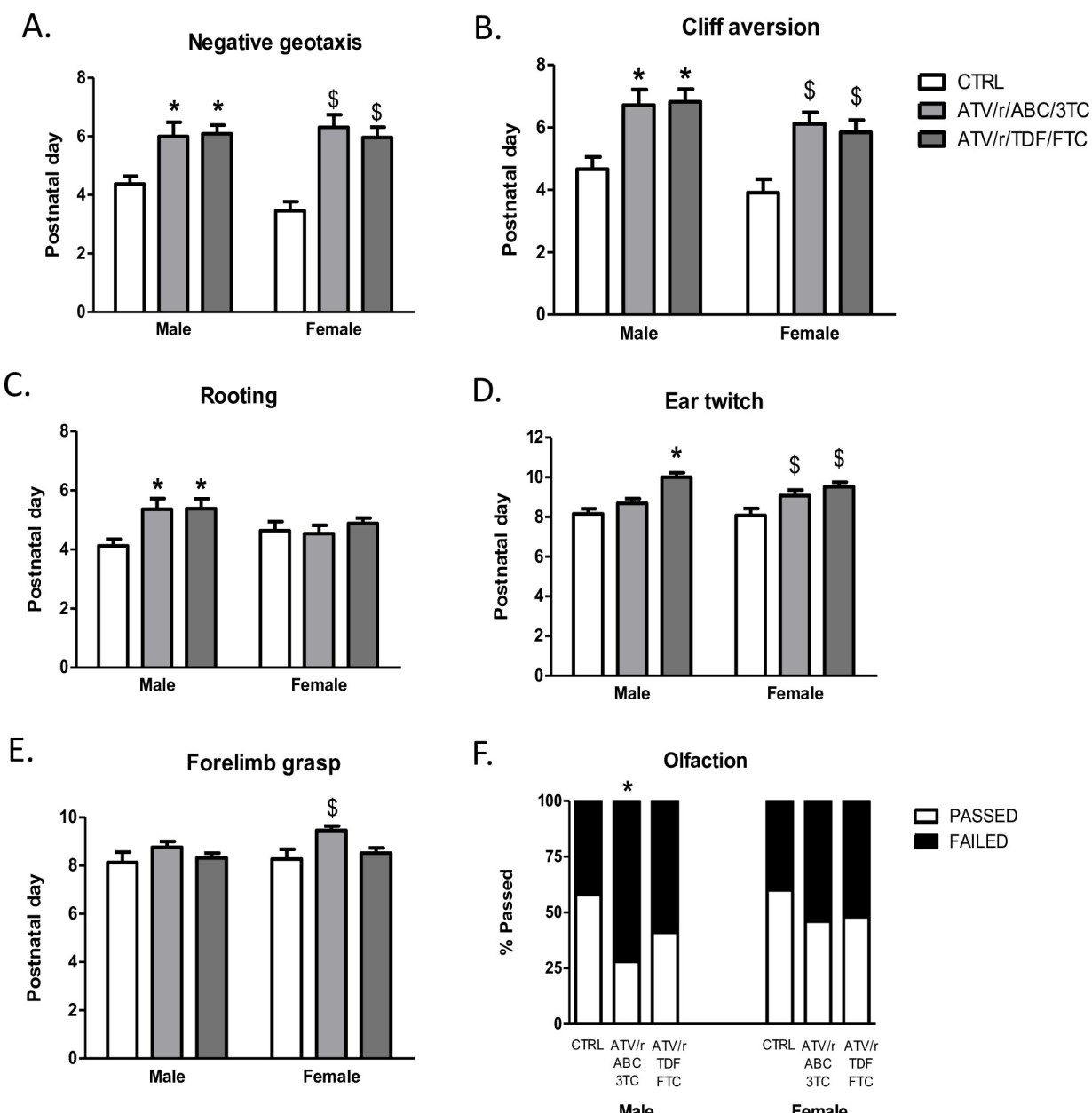

**Fig 4. In-utero exposure to ART delays development of certain primitive reflexes.** Postnatal day of successful acquisition of negative geotaxis reflex (A), cliff aversion reflex (B), rooting reflex (C), ear twitch reflex (D), and forelimb grasp reflex (E) for male and female pups exposed in-utero to control (CTRL, white bars), ATV/r/ABC/3TC (light grey bars), or ATV/r/TDF/FTC (dark grey bars). Data are mean ± SEM (n = 11–34 pups/group). A mixed effects model was used to examine differences between control and treatment arms (fixed effect) accounting for litter effects (random effect) stratified by sex. *p<0.05 compared to control males. $p<0.05 compared to control females. Percentage of pups that successfully completed or failed the olfaction homing test on postnatal day 11 is shown in (F). * p<0.05 vs. control by chi-squared test. ATV/r, ritonavir-boosted atazanavir; ABC, abacavir; 3TC, lamivudine; TDF, tenofovir; FTC, emtricitabine; CTRL, control.

Taken together, our results show that U-ART regimens induced delays in the development of certain primitive reflexes, especially the ones that are reflective of sensory-motor, tactile, and neuro-muscular developments of the nervous system. A summary of the findings is shown in Table 1.

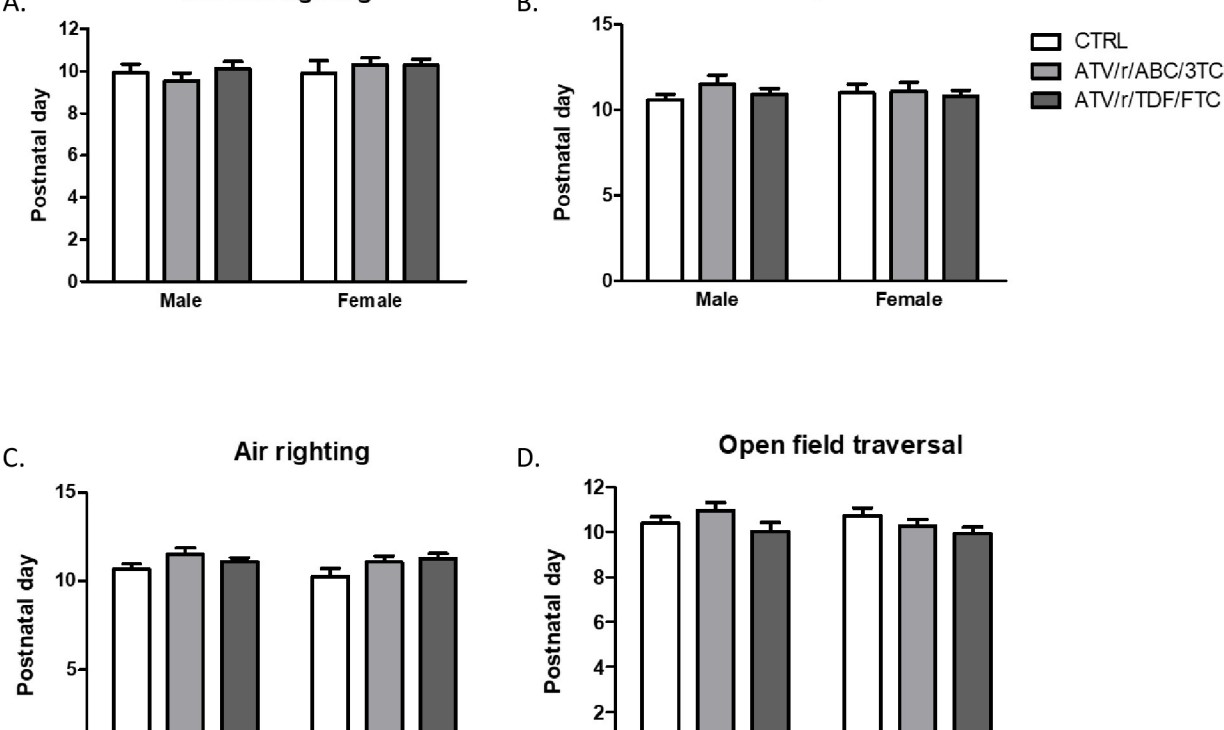

**Fig 5. In-utero exposure to ART does not disrupt the development of every primitive reflex.** Postnatal day of successful acquisition of surface righting reflex (A), auditory reflex (B), air righting reflex (C), and open field traversal (D) for male and female pups exposed in-utero to control (CTRL, white bars), ATV/r/ABC/3TC (light grey bars), or ATV/r/TDF/FTC (dark grey bars). Data are mean ± SEM (n = 11–34 pups/group). A mixed effects model was used to examine differences between control and treatment arms (fixed effect) accounting for litter effects (random effect) stratified by sex. No statistically significant differences were observed. ATV/r, ritonavir-boosted atazanavir; ABC, abacavir; 3TC, lamivudine; TDF, tenofovir; FTC, emtricitabine; CTRL, control.

## Behavioural performance in adulthood on the OFT

To investigate the consequences of delayed milestones development in adulthood, we tested mice for general activity, exploration, speed of locomotion, and anxiety-like behavior in the OFT. Our results showed no significant differences between treatment arms for speed, total distance moved, distance and ambulatory time in the centre and the peripheral zones of the arena, and rearing and resting (Fig 6A–6H) in males. In females, no significant differences were noted between treatment groups for speed of locomotion (Fig 6A) and rearing (vertical counts) (Fig 6G). Differences were observed in total distance moved in the entire arena (p = 0.03) (Fig 6B), in the centre (p = 0.06) (Fig 6C) and in the periphery (p = 0.04) (Fig 6D), ambulatory time spent in the centre (p = 0.03) (Fig 6E) and in periphery (p = 0.07) (Fig 6F) for females. After correcting for multiple comparisons, the ATV/r/TDF/FTC exposed females demonstrated significantly lower total distance travelled and less ambulatory time in the centre compared to controls. Differences were also observed in resting time in females (p = 0.03, Fig 6H), with significantly higher resting times observed in the ATV/r/ABC/3TC exposed females compared to controls.

## Discussion

We have shown here for the first time that gestational exposure to PI-based antiretroviral regimens delays somatic growth and the appearance of primitive reflexes in mouse pups. Pups of

**Table 1. Summary of differences in developmental milestone findings for each ART regimen compared to control.**

| Days delay vs. control | ATV/r/ABC/3TC | | ATV/r/TDF/FTC | |
|---|---|---|---|---|
| | **Male** | **Female** | **Male** | **Female** |
| Negative geotaxis | 2 days delay | 2.5 days delay | 2 days delay | 2.5 days delay |
| Cliff aversion | 2 days delay | 2 days delay | 2 days delay | 2 days delay |
| Rooting | 1 day delay | No delay | 1 day delay | No delay |
| Outer ear opening | No delay | 0.5 day delay | 0.5 day delay | 0.5 day delay |
| Ear twitch | No delay | 1 day delay | 1.5 days delay | 1.5 days delay |
| Forelimb grasp | No delay | 1 day delay | No delay | No delay |
| Surface righting | No delay | No delay | No delay | No delay |
| Air righting | No delay | No delay | No delay | No delay |
| Auditory reflex | No delay | No delay | No delay | No delay |
| Open field traversal | No delay | No delay | No delay | No delay |
| Eyes opening | No delay | No delay | No delay | No delay |
| Incisors eruption | No delay | No delay | No delay | No delay |
| Fur appearance | No delay | No delay | No delay | No delay |
| Olfaction | Deficit present | No deficit | No deficit | No deficit |

ATV/r, ritonavir-boosted atazanavir; ABC, abacavir; 3TC, lamivudine; TDF, tenofovir; FTC, emtricitabine.

either sex, exposed to ATV/r/ABC/3TC, and ATV/r/ TDF/FTC exhibited lower birth weights, body lengths, and delays in the appearance of negative-geotaxis and cliff-aversion reflexes. Exposure to either regimen delayed the appearance of rooting reflex in the male pups. Exposure to ATV/r/ABC/3TC alone delayed appearance of the olfactory reflex in males and forelimb grasp in female pups. Exposure to ATV/r/TDF/FTC delayed outer ear detachment and ear-twitch response in both sexes. Surface-righting, air-righting, auditory-reflex, and open field traversal were unaffected by either regimen. The appearance of fur, eyes, and incisors were also not affected by *in utero* exposure to either regimen.

In our study, we treated pregnant dams with clinically relevant doses of ATV/r/ABC/3TC and ATV/r/TDF/FTC from GD 0.5 till delivery; a time window that approximately corresponds to gestational weeks (GW) 1–24 in humans [28, 29]. Perturbations of the *in utero* environment during the first trimester of pregnancy are known to have life-long impacts on offspring brain and behavior [31]. First-trimester exposure to PI-based ARTs have indeed been associated with higher incidences of abnormalities related to growth [32], sensory [33] and neurocognitive behaviors [34] in CHEUs.

Key developmental events that mark this critical period are conserved in mammals. This allows translation of early neurodevelopmental milestones between species (Fig 1). For instance, developmental milestones such as neurulation begins on GD9–9.5 in mice and GD24-28 in humans. Neurogenesis, gliogenesis, synaptogenesis, and myelination continue postnatally in both species. At P10 the rodent brain is comparable to that of a term infant and by P20, it corresponds to that of a human, 2–3 years old [28]. Certain developmental processes such as maturation of specific neuronal regions can be temporally correlated to the appearance / disappearance of primitive reflexes in neonates [35]. For example, cortical inhibition of the brain stem around 3 months of age is known to correspond to the disappearance of palmer grasp in babies, a reflex similar to forelimb grasp in rodents and non-human primates [36, 37]. Owing to the conservation of developmental events, the age at which certain behaviors occur in rodents can be mapped onto that of humans.

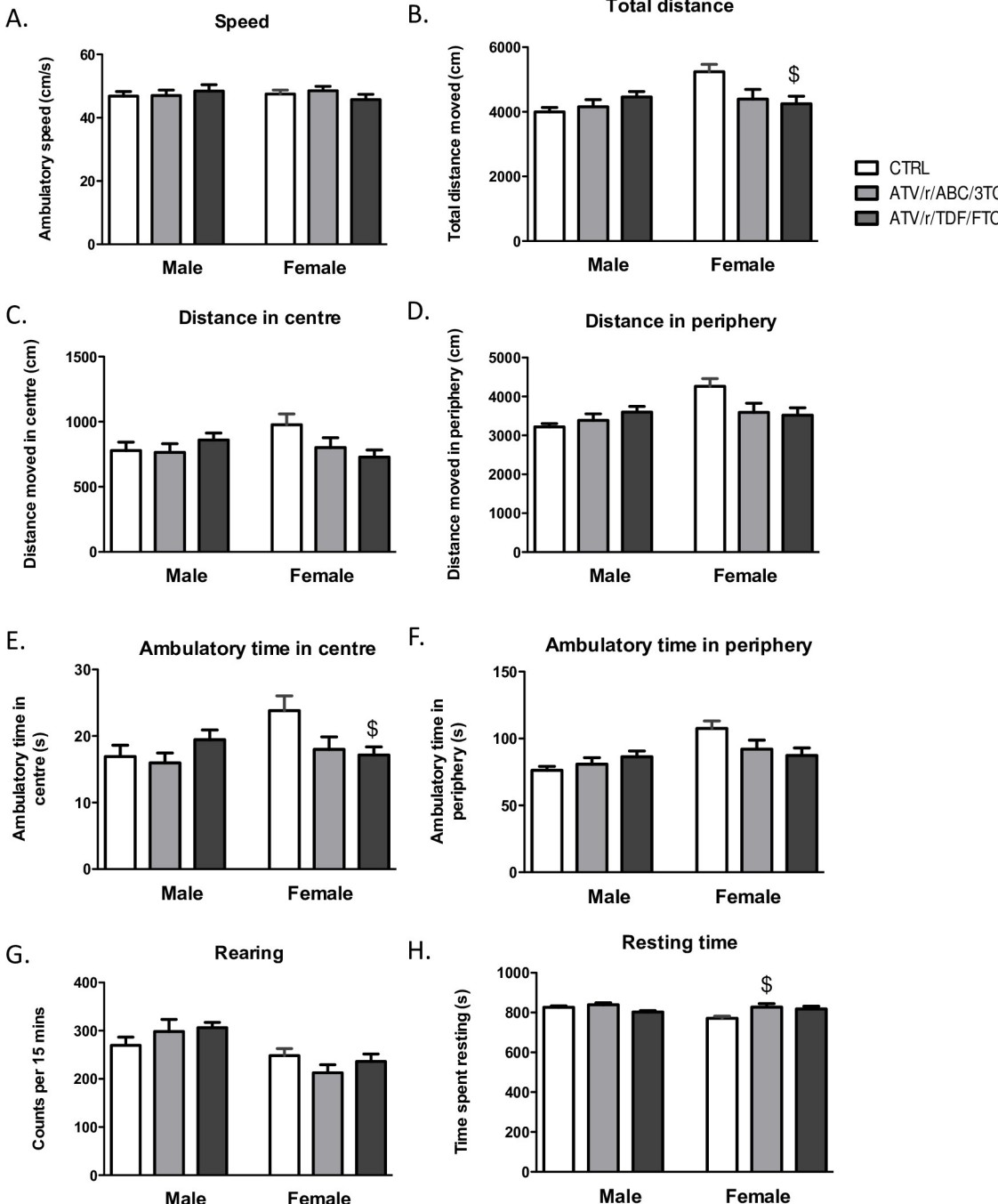

**Fig 6. In-utero exposure to ART alters behaviour in the open field test in adulthood.** Speed of locomotion (A), total distance moved (B), distance moved in the centre of the arena (C), distance moved in the periphery (D), ambulatory time in the centre of the arena (E), ambulatory time in the periphery (F), rearing time (G), and resting time (H) for male and female mice exposed in-utero to control (CTRL, white bars), ATV/r/ABC/3TC (light grey bars), or ATV/r/TDF/FTC (dark grey bars). Data are mean ± SEM (n = 12–17 mice/group). One-way ANOVA with Bonferroni multiple comparisons tests was used to examine differences between control and treatment arms for each sex. $ indicates p<0.05 compared to control females. ATV/r, ritonavir-boosted atazanavir; ABC, abacavir; 3TC, lamivudine; TDF, tenofovir; FTC, emtricitabine; CTRL, control.

Mice exposed to ATV/r/ABC/3TC and ATV/r/ TDF/FTC were born with significantly lower birth weight and length compared to controls. These signs of retarded somatic growth,

however, lasted only till P3. Our results corroborate clinical findings that have shown HEU infants to have significantly lower body weights and lengths at birth [3, 4], with catch up growth by 6 months of age [4], although some studies have shown persistent growth defects in CHEUs [31]. Differences in the type of ART exposure and socioeconomic factors may explain these differences. Slower postnatal growth was also seen previously in CD1 mice exposed to AZT and 3TC [38]. Low birth weight or being born small for gestational age (SGA) may be indicators of altered *in utero* environments and/or placental insufficiencies [39]. Maternal treatment with ARTs could influence the fetus directly (mitochondrial and neuronal toxicity) [40, 41], or indirectly (altered maternal hormonal systems and placental environment) [42] and thereby regulate growth. Growth restricted and pre-term infants are known to have higher predispositions for neurobehavioral and intellectual disabilities related to learning, cognition, and socialization [43]. CHEUs, having higher incidences of SGA, are also known to have worse neurocognitive outcomes when compared to unexposed children [3, 44].

A small dip in the body weights of control pups, both males and females were observed transiently at P3, although a concomitant decline in body length was not observed in these pups. It is possible that the transient weight loss (thinning, but not stunting) may have occurred due to insufficient feeding/nursing/hydration of these pups on that day.

Our data suggest that reflex ontogeny is disrupted in pups exposed *in utero* to ATV/r/ABC/ 3TC and ATV/r/TDF/FTC. We assessed sensory-motor reflex development in our pups by testing for surface-righting, air-righting, negative-geotaxis, and cliff-aversion. We observed delays in negative-geotaxis and cliff-aversion with U-ART, but no delays in the righting reflexes. A mature response in all four of these tests requires coordination between the vestibular system, muscles, and spine, but surface-righting, negative-geotaxis, and cliff-aversion additionally require inputs of tactile sensations via whiskers to the somatosensory cortex [45]. For surface-righting, pups utilize sensory information from proprioceptive-tactile sources as well as from vestibular sources. For air-righting on the other hand, pups are dependent primarily on their vestibular systems [46]. The difference in timing of maturation of surface (P8) versus air-righting (P11) reflexes is reflective of the relative difference in the maturation of tactile versus the vestibular systems [46]. Both types of righting reflexes, negative-geotaxis and cliff-aversion, require the concurrent development of strength and coordination, and a delay in appearance of these reflexes could also indicate retarded muscular development and growth. Our results showed no effect of U-ART on either type of righting reflex. Interestingly, both U-ART regimens delayed the appearance of negative-geotaxis and cliff-aversion; reflexes that are dependent partly on tactile inputs to the somatosensory neural pathways [45].

Somatosensory pathway maturation was also assessed via testing for rooting reflex. Rooting, a response to tactile stimulation of the skin and whiskers on the snout was delayed by both regimens, albeit only in the male pups. Although it is difficult to pinpoint why the regimens affected the male offspring selectively for this reflex, one may speculate that the regimens impacted the underlying neurocircuitry differently in the two sexes, by neurohormonal interferences at a critical developmental phase. Given that male placentas are more responsive to changes in the maternal environment, such as maternal exposure to drugs [47], maternal ART may have impacted the male fetuses differently from the females. Previous studies suggest sex differences for the effects of gestational exposure to other NRTI regimens. *In utero* exposure of CD1 mice to AZT [48], 3TC [49] and AZT+3TC [38] from GD10 to GD21 delayed emergence of pole grasp reflex in males [48] and was associated with deficits in social interaction only in females at P35 [38]. Corroborating these findings, we show here a delay in the emergence of forelimb grasp selectively in female pups exposed to ATV/r/ABC/3TC treatment. Differences between ours and previous findings could be due to differences in drug regimens, duration of treatment, and strain of mice used. Collectively, the sex divergent effects of various U-ART

regimens on several milestone tests strengthens the idea that U-ARTs can affect offspring neu-rodevelopment based on offspring sex.

Further, we show that ATV/r/ABC/3TC exposure impairs olfaction in male pups in the homing test. Failure in neonatal homing test is a hallmark for olfactory deficits in rodent mod-els of autism spectrum disorder [50, 51]. In fact, impaired olfaction is considered a vulnerabil-ity marker for several psychiatric disorders including autism, schizophrenia and depression [52]. Because of the vast network and direct connectivity of the olfactory tract to various brain regions including the entorhinal cortex, the hippocampus, and the amygdala, a variety of vital activities such as feeding, reproduction, social behavior and memory rely on the sense of smell and are intricately interlinked. The olfactory system is also closely linked to the immune sys-tem [53]. An early impairment in olfaction could be a tell-tale sign for more deficits in other neuro-behavioral areas in the future. A significant increase in the odds for an autism spectrum diagnosis in CHEUs compared to unexposed children was reported in a retrospective study in British Columbia Canada, although an association with U-ART exposure was not observed [54].

ATV/r/ABC/3TC and ATV/r/TDF/FTC had mutually exclusive effects on some of the mile-stones tested. For example, concurrent delays in the opening of the outer ear and the emer-gence of ear-twitch responses were observed in the ATV/r/TDF/FTC group only. These effects were not observed in the ATV/r/ABC/3TC exposed group. Since ATV/r was common to both regimens, we speculate that differences in the effects of the two regimens may have risen due to differences in the composition of the NRTI backbones. Although ABC, 3TC, TDF, and FTC are all nucleos(t)ide analogs, they are different molecular entities. These viral reverse transcrip-tase inhibitors are known to interrupt mtDNA replication through host mitochondrial DNA-polymerase-γ (mtDNA-pol-γ) binding. Most of the clinical manifestations of NRTI toxicities resemble mitochondrial diseases resulting from mutations in mtDNA [55, 56]. Neurological disorders associated with mitochondrial dysfunction have been reported in CHEUs exposed to NRTIs [57–60]. Differences in pharmacokinetics and the binding affinities of the different NRTIs to mtDNA-pol-γ [61] may explain the differences observed in their effects on the mile-stones tested.

We further showed that consequences of prenatal exposure to ART persist beyond postnatal ages, into adulthood. Exposure to ATV/r/TDF/FTC lowered the levels of exploration and activity in the open field in female mice. Resting or idle time was also found to be higher in females exposed to ATV/r/ABC/3TC. Although not anxiety-like, the behavioral phenotype evi-dent in U-ART females was one of lowered overall exploration of the arena and increased rest-ing/ idling. Decreased exploration in the OFT has been linked to abnormalities in neurotransmitter systems that regulate locomotion [62, 63], motivation and stress response [64], such as the dopaminergic, cholinergic and serotonergic systems and brain areas like the hippocampus [64], striatum and cortex [62, 63]. One previous study, testing AZT+3TC exposed CD1 mice on OFT in adulthood, had shown increased rearing and no changes in gen-eral locomotion [65]. Our results show that U-ART is not without consequences on behavior in adulthood, although the observed defects are minor. Further investigation into the long-term effects of U-ART on brain and behavior is merited.

There are several strengths to our study. We have identified the adverse effects of U-ART on early development, in the absence of clinical and socioeconomic confounding factors. Drug levels used in our study, mimic levels seen in pregnant women. Considering sex as a variable in all our investigations, we have highlighted the sex divergent effects of U-ART. The limita-tions of our study include the lack of HIV infection in the dams, which is not reflective of the clinical scenario. HIV infection can also alter the *in utero* environment and thus likely modifies the effects on perinatal ART exposure in CHEUs. Further, in our model, we only expose the

pups to ART *in utero*, a time window that maps to GW1-24 in humans. While this time period encompasses fetal neurogenesis and migration it does not encompass the degree of maturation that occurs in human pregnancy. Thus, our results may underestimate the clinical impact.

## Conclusions

The major novel finding in our study is that mice exposed *in utero* to ATV/r/ABC/3TC and ATV/r/TDF/FTC exhibit delays in the development of milestones, particularly in reflexes that require the ability for dynamic postural adjustments, muscular strength and are dependent on whisker based tactile stimulation of somatosensory pathways. These findings are potentially indicative of disrupted underlying neurocircuitry. The rate of maturation in mice during the first postnatal month being 150 times that of humans [66], our work suggests that even minor delays in mouse reflex developments, when extrapolated onto the human timescale, could translate into weeks or months of delay. Clinical studies have associated *in utero* exposure of AZT, 3TC, and nevirapine to retention (delayed disappearance) of primitive reflexes in CHEUs by several months [10]. Our work, therefore, highlights the need for investigating the persistence and the etiology of the effects of U-ART, for better understanding of the spectrum of health issues that may affect CHEUs, and for better optimization of perinatal anti-HIV regimens.

## Supporting information

**S1 Fig. Example of a data sheet for recording observations for the assessments of developmental milestones.** The shaded boxes mark the tests that were performed on each day. Tests were done in a sequence that was age appropriate for mice pups.
(TIF)

**S1 Data. Complete raw data set.**
(XLSX)

## Author Contributions

**Conceptualization:** Ambalika Sarkar, Kayode Balogun, Howard T. Mount, Lena Serghides.

**Formal analysis:** Ambalika Sarkar, Kayode Balogun, Monica S. Guzman Lenis, Lena Serghides.

**Funding acquisition:** Lena Serghides.

**Investigation:** Ambalika Sarkar, Kayode Balogun, Sebastian Acosta.

**Project administration:** Monica S. Guzman Lenis.

**Supervision:** Lena Serghides.

**Writing – original draft:** Ambalika Sarkar.

**Writing – review & editing:** Kayode Balogun, Monica S. Guzman Lenis, Sebastian Acosta, Howard T. Mount, Lena Serghides.

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
