## [Decision Letter · Decision Letter 0]

2 Sep 2020

PONE-D-20-20335

In utero exposure to protease inhibitor-based antiretroviral regimens disrupts growth and developmental milestones in mice

PLOS ONE

Dear Dr. Serghides,

Thank you for submitting your manuscript to PLOS ONE. After careful consideration, we feel that it has merit but does not fully meet PLOS ONE’s publication criteria as it currently stands. Therefore, we invite you to submit a revised version of the manuscript that addresses all the  points raised during the review process. Especially, you should strengthen your data by extending the array of tests on the developmental, behavior and molecular effects of U-ART treatment.

We look forward to receiving your revised manuscript.

Kind regards,

Jean-Léon Thomas

Academic Editor

PLOS ONE

Journal Requirements:

2. To comply with PLOS ONE submissions requirements, please provide methods of sacrifice in the Methods section of your manuscript.

4.Thank you for stating the following in the Competing Interests section:

[The authors have no conflicts of interest to disclose in relation to this work.  LS reports personal fees from ViiV Healthcare for participation in a Women and Transgender Think Tank. ].

Reviewers' comments:

Reviewer's Responses to Questions

**Comments to the Author**

1. Is the manuscript technically sound, and do the data support the conclusions?

Reviewer #1: Yes

Reviewer #2: Partly

2. Has the statistical analysis been performed appropriately and rigorously? 

Reviewer #1: Yes

Reviewer #2: Yes

3. Have the authors made all data underlying the findings in their manuscript fully available?

Reviewer #1: Yes

Reviewer #2: Yes

4. Is the manuscript presented in an intelligible fashion and written in standard English?

Reviewer #1: Yes

Reviewer #2: Yes

5. Review Comments to the Author

Reviewer #1: In this study, Sarkar et al demonstrated for the first time using a mouse model that an antiretroviral therapy (ART), which is commonly used in pregnant women to prevent the vertical transmission of HIV-1, delays growth and neurocognitive development of the offspring, as has been suggested in humans. Therefore, this study may serve for understanding the underlying mechanisms for the adverse effects and further optimization of the ART regimens during pregnancy. The manuscript is written quite clearly and comprehensively. I have only a few suggestions.

1. In Fig. 2A and C, the body weight of both control male and female mice decreased from day 2 to day3 post-birth, although the body lengths (Fig. 2E and F) kept growing during the period. This looks unnatural for me and probably also for the readers. It is better to clarify in the manuscript whether this is commonly observed in normal pups and if not, better to address/suggest why it has happened.

2. There are some sentences in the text that “no difference (differences) was (were) observed between …”. It would be better to add words such as no “significant” difference or no “remarkable” difference etc because there are more or less differences between the groups in the data.

Reviewer #2: Sarkar et al. demonstrated that U-ART treatment delayed the development of pups via mouse model. In this manuscript, the authors utilized various development indicator to monitor the pups growth after U-ART. The results showed that the somatic development and certain primitive reflexes seemed delayed in U-ART pups. (Fig.2 &Fig.4).

Some questions were raised here after reviewing these results,

1. According to the results shown in Fig. 2-3, U-ART pups seemed to reach the average body weight and other somatic development before weaning. Did these results reflect the fact in human cases? Because the authors used “disrupt” in the title, the results did not show a significant disruption of development, for example, somatic defect, what is the fact in the human cases? Development delayed or some defects were observed in the U-ADT infants?

2. Because the authors mentioned that the mortality was higher in the first year in human cases, did the authors try to count the viability of pups between control and U-ART groups?

3. Since the development indicators cannot reflect too many differences in control and U-ART pups, did the authors try to infect with pathogens or bacterial stimulators (e.g LPS) to see if the U-ART groups were more susceptible to infections?

4. If the authors extend the monitoring window after weaning and more examinations can be utilized in the experiment, such as behavior tests, is it possible to see more differences between control and U-ART groups?

5. The author mentioned that the mtDNA is affected because of ART treatment, is it possible to do more investigation in mtDNA in U-ART pups?

Although the novelty of this manuscript is quite unique, the data here seems to be descriptive and preliminary.

6. PLOS authors have the option to publish the peer review history of their article (what does this mean?). If published, this will include your full peer review and any attached files.

Reviewer #1: No

Reviewer #2: No

---

## [Author Response · Author response to Decision Letter 0]

19 Oct 2020

Responses to Reviewers:

Reviewer #1: In this study, Sarkar et al demonstrated for the first time using a mouse model that an antiretroviral therapy (ART), which is commonly used in pregnant women to prevent the vertical transmission of HIV-1, delays growth and neurocognitive development of the offspring, as has been suggested in humans. Therefore, this study may serve for understanding the underlying mechanisms for the adverse effects and further optimization of the ART regimens during pregnancy. The manuscript is written quite clearly and comprehensively. I have only a

few suggestions.

1. In Fig. 2A and C, the body weight of both control male and female mice decreased from day 2 to day3 post-birth, although the body lengths (Fig. 2E and F) kept growing during the period. This looks unnatural for me and probably also for the readers. It is better to clarify in the manuscript whether this is commonly observed in normal pups and if not, better to address / suggest why it has happened.

In response to the reviewer’s comment we re-examined the body weight data for all pups. The decline in weight seen on day 3 in the controls was driven by 2 litters, which lost weight on day 3. This weight loss was transient and the pups recovered in weight on subsequent days. 

Literature supports strong correlations between pup’s body weight and dam’s lactation, dam’s food intake, home cage temperature and relative humidity [1, 2]. These papers show that increases in body heat generation in lactating dams, and increase in relative humidity in cage, lowers food intake transiently which corresponds to transient dips in pup body weights. This happens in spurts across the postnatal period, with the first spurt being around P3 [2]. It is most likely that the dip in weight visible in our graph at P3 in our control group, with no concomitant decrease seen in body lengths, is merely an aberration, which could have been caused due to a transient decrease in nursing / feeding / hydration of the pups by the two dams.

We thank the reviewer for raising this question. It gave us a chance to clarify this issue in our manuscript. We have now added an explanation in the revised Discussion page 19 line 389-392.

2. There are some sentences in the text that ‘no difference (differences) was (were) observed between…’. It would be better to add words such as no ‘significant’ difference or no remarkable difference’ etc because there are more or less differences between the groups in the data.

We have now changed our sentences from ‘no differences were observed’ to ‘no remarkable or significant differences were observed’ wherever applicable, as per the reviewer’s recommendation. We have made these changes under ‘Results’ section lines 216, 236, and 294.

Reviewer #2: Sarkar et al. demonstrated that U-ART treatment delayed the development of pups via mouse model. In this manuscript, the authors utilized various development indicator to monitor the pups growth after U-ART. The results showed that the somatic development and certain primitive reflexes seemed delayed in U-ART pups. (Fig.2 & Fig.4).

Some questions were raised here after reviewing these results,

1. According to the results shown in Fig. 2-3, U-ART pups seemed to reach the average body weight and other somatic development before weaning. Did these results reflect the fact in human cases? 

Several studies have reported delayed increase in body weight and other somatic developments in infants that are HEU (Reviewed in Wedderburn et al, 2019 [3]). Most studies that followed HEU children for the first 2 years of their lives have reported delayed somatic developments such as: lower birth weight [4], lower weight-for-age Z scores [5], lower length-for-age Z (LAZ) scores [6], three-fold higher incidences of stunting [7, 8], and higher incidences of microcephaly (head circumference-for-age Z scores) [5, 9]. While most studies have shown somatic growth delays to continue till about 2 years of life, the Powis et al study from Botswana had shown lower WAZ amongst HEU infants improved rapidly in the first 2 months of life and reached the expected average-for-age values by 6 months of age. 

Similar to some of the findings listed above, especially Powis et al, our study in mice also shows that U-ART pups reach average body weight and other somatic developments before the time of weaning. 

Another potentially interesting concurrence between our data and that from HEU studies is that it took longer for U-ART pups (particularly in female pups) to catch up in length to the controls, as compared to weight. This coincides with evidence of lower LAZ persisting at 6 months [8] and 2 years of age for HEU children [6]. 

Because the authors used ‘disrupt’ in the title, the results did not show a significant disruption of development, for example, somatic defect, what is the fact in the human cases? Development delayed or some defects were observed in the U-ART infants?

We agree with the reviewer that the phenotype we are reporting here is more of ‘delayed growth and delayed development’, rather than one of ‘permanent disruption or defect’. Therefore, we have now removed the word ‘disrupts’ from the title of our paper and replaced it with ‘delays growth and developmental milestones’. 

2. Because the authors mentioned that the mortality was higher in the first year in human cases, did the authors try to count the viability of pups between control and U-ART groups?

We thank the reviewer for raising this question. We did record mortality of pups in each of the treatment arms on the day of birth (P0) and throughout the course of the experiments. Of the 42 pups born in the control group, 0 were found dead on P0 (0% mortality). Of the 66 pups born in the KVX group, 2 were found to be dead (3% mortality). Of the 78 pups born in the TRV group, 5 were found to be dead (6.4% mortality) on P0. No mortality was observed after P0. The difference in mortality between groups did not reach significance (chi-squared test, p=0.14). We have added these data to the Results section of the revised paper (line 195-202).

Mortality amongst HEUs has been shown to be higher in resource limited settings, especially in cases where confounding factors such poor socio-economic status, malnutrition and unhygienic living conditions exist. HEUs are known to have a weaker immune system and respond poorly to infectious diseases [10]. It is possible that we did not observe significantly higher mortality in U-ART pups because of the controlled and pathogen free laboratory conditions in which they were raised.

3. Since the development indicators cannot reflect too many differences in control and U-ART pups, did the authors try to infect with pathogens or bacterial stimulators (e.g LPS) to see if the U-ART groups were more susceptible to infections?

We thank the reviewer for suggesting this very interesting experiment. However, in this paper we have tried to showcase only the early postnatal growth and behavioral characteristics (reflex maturation) of U-ART pups. We tried to concentrate on the neurodevelopmental delays that ensue in-utero ART exposure, which could be indicators for more robust behavioral abnormalities, later in life. Infecting U-ART pups with pathogens or bacterial stimulators to monitor their immune responses, susceptibility to infections, mortality rates etc, would be a deviation from the objectives of this paper. Experiments along the lines suggested by the reviewer are some of the future directions of our project and are beyond the scope of the current paper.

4. If the authors extend the monitoring window after weaning and more examinations can be utilized in the experiment, such as behavior tests, is it possible to see more differences between control and U-ART groups?

To address this question, we extended our study and performed additional behavioral experiments on our animals in adulthood. We performed the open field test to investigate general activity, locomotion and anxiety-like behavior in U-ART animals at 2 months of age. As has been pointed out by the reviewer, we did see more differences between the control and U-ART animals at this age. These new data have been added to the revised manuscript – see Figure 6, Methods line 153-160, Results line 318-331, and Discussion line 453-465 of the revised manuscript. 

Our results showed that exposure to ATV/r/TDF/FTC had an effect on exploration levels in the open field in female mice (Figure 6), a decline in total distance moved in the open filed and distance moved in the centre. While a decline in total distance moved in the entire arena does not signify anxiety-like behavior, it indicates a decline in locomotion / motivation for exploration overall. A significant decline in ambulatory time spent in the centre was also noted in ATV/r/ TDF/FTC exposed females. Resting or idle time was also found to be higher in females exposed to ATV/r/ABC/3TC. Similar to our results from the post-natal period, we observed a sex difference in the effects of U-ART once again in adulthood. In males, no significant differences were observed in any of the measures in either U-ART exposed arms, compared to controls. We have discussed possible reasons for observation of sex differences in effects of U-ART under our revised Discussion. 

5. The author mentioned that the mtDNA is affected because of ART treatment, is it possible to do more investigation in mtDNA in U-ART pups?

The effect of ART on mtDNA is well investigated and has been reviewed extensively [11]. It is possible to carry out several experiments to check for the effects of ART on mtDNA in U-ART pups. mtDNA integrity and mtDNA transcript levels can be checked for in mitochondrial lysates obtained from control versus U-ART mice tissue. However, such experiments would require careful and purposeful planning and collection of tissue in a systematic way. Unfortunately, this is not possible with our current experimental setup. We thank the reviewer for raising this question and it certainly provides impetus for future investigations. 

Although the novelty of this manuscript is quite unique, the data here seems to be descriptive and preliminary.

We agree with the reviewer that our study is descriptive. However, our experiments include a sufficient number of animals, we have examined a large number of developmental milestones and somatic characteristics, and we have performed analyses based on pup sex. We have now, following the reviewer’s excellent suggestion, also expanded to include a behavioural test in adulthood. While we agree that our data are not mechanistic but rather descriptive, we respectfully disagree that they are preliminary.

References:

1. Król E, Murphy M, Speakman JR. Limits to sustained energy intake. X. Effects of fur removal on reproductive performance in laboratory mice. J Exp Biol. 2007;210(Pt 23):4233-4243. doi:10.1242/jeb.009779

2. Spangenberg E, Wallenbeck A, Eklöf AC, Carlstedt-Duke J, Tjäder S. Housing breeding mice in three different IVC systems: maternal performance and pup development. Lab Anim. 2014;48(3):193-206. doi:10.1177/0023677214531569

3. Wedderburn CJ, Evans C, Yeung S, Gibb DM, Donald KA, Prendergast AJ. Growth and Neurodevelopment of HIV-Exposed Uninfected Children: a Conceptual Framework. Curr HIV/AIDS Rep. 2019;16(6):501-513. doi:10.1007/s11904-019-00459-0

4. Dara JS, Hanna DB, Anastos K, Wright R, Herold BC. Low Birth Weight in Human Immunodeficiency Virus-Exposed Uninfected Infants in Bronx, New York. J Pediatric Infect Dis Soc. 2018;7(2):e24-e29. doi:10.1093/jpids/pix111

5. Evans C, Chasekwa B, Ntozini R, Majo FD, Mutasa K, Tavengwa N, et al. Surviving and thriving? Outcomes of HIV-exposed children in rural Zimbabwe. Conference on Retroviruses and Opportunistic Infections (CROI); 4th–7th March; Seattle (WA)2019

6. le Roux SM, Abrams EJ, Donald KA, Brittain K, Phillips TK, Nguyen KK, et al. Growth trajectories of breastfed HIV-exposed uninfected and HIV-unexposed children under conditions of universal maternal antiretroviral therapy: a prospective study. Lancet Child Adolesc Health. 2019;3(4):234–44. 

7. Sudfeld CR, Lei Q, Chinyanga Y, et al. Linear Growth Faltering Among HIV-Exposed Uninfected Children. J Acquir Immune Defic Syndr. 2016;73(2):182-189. doi:10.1097/QAI.0000000000001034

8. Powis KM, Smeaton L, Ogwu A, et al. Effects of in utero antiretroviral exposure on longitudinal growth of HIV-exposed uninfected infants in Botswana. J Acquir Immune Defic Syndr. 2011;56:131–8. 

9. Spaulding AB, Yu Q, Civitello L, et al. Neurologic Outcomes in HIV-Exposed/Uninfected Infants Exposed to Antiretroviral Drugs During Pregnancy in Latin America and the Caribbean. AIDS Res Hum Retroviruses. 2016;32(4):349-356. doi:10.1089/AID.2015.0254

10. Afran L, Garcia Knight M, Nduati E, Urban BC, Heyderman RS, Rowland-Jones SL. HIV-exposed uninfected children: a growing population with a vulnerable immune system. Clin Exp Immunol. 2014;176(1):11‐22. doi:10.1111/cei.12251

11. Pinti M, Salomoni P, Cossarizza A. Anti-HIV drugs and the mitochondria. Biochim Biophys Acta. 2006;1757(5-6):700-707. doi:10.1016/j.bbabio.2006.05.001

---

## [Decision Letter · Decision Letter 1]

4 Nov 2020

In utero exposure to protease inhibitor-based antiretroviral regimens delays growth and developmental milestones in mice

PONE-D-20-20335R1

Dear Dr. Serghides,

We’re pleased to inform you that your manuscript has been judged scientifically suitable for publication and will be formally accepted for publication once it meets all outstanding technical requirements.

Kind regards,

Jean-Léon Thomas

Academic Editor

PLOS ONE

Additional Editor Comments (optional):

Reviewers' comments:

Reviewer's Responses to Questions

**Comments to the Author**

1. If the authors have adequately addressed your comments raised in a previous round of review and you feel that this manuscript is now acceptable for publication, you may indicate that here to bypass the “Comments to the Author” section, enter your conflict of interest statement in the “Confidential to Editor” section, and submit your "Accept" recommendation.

Reviewer #1: All comments have been addressed

Reviewer #2: All comments have been addressed

2. Is the manuscript technically sound, and do the data support the conclusions?

Reviewer #1: Yes

Reviewer #2: Yes

3. Has the statistical analysis been performed appropriately and rigorously? 

Reviewer #1: Yes

Reviewer #2: Yes

4. Have the authors made all data underlying the findings in their manuscript fully available?

Reviewer #1: Yes

Reviewer #2: Yes

5. Is the manuscript presented in an intelligible fashion and written in standard English?

Reviewer #1: Yes

Reviewer #2: Yes

6. Review Comments to the Author

Reviewer #1: The authors have responded well to each of the Reviewers' comments and improved the manuscript accordingly. There are no further comments from this Reviewer.

Reviewer #2: The authors answered all the comment point to point and also did more experiments to strengthen the data in the manuscript.

7. PLOS authors have the option to publish the peer review history of their article (what does this mean?). If published, this will include your full peer review and any attached files.

Reviewer #1: No

Reviewer #2: No

---

## [Editor Report · Acceptance letter]

10 Nov 2020

PONE-D-20-20335R1 

*In utero* exposure to protease inhibitor-based antiretroviral regimens delays growth and developmental milestones in mice 

Dear Dr. Serghides:

I'm pleased to inform you that your manuscript has been deemed suitable for publication in PLOS ONE. Congratulations! Your manuscript is now with our production department. 

Kind regards, 

on behalf of

Dr. Jean-Léon Thomas 

Academic Editor

PLOS ONE